# Genetic validation of *Pf*FKBP35 as an antimalarial drug target

Basil T Thommen[1,2], Jerzy M Dziekan[3], Fiona Achcar[4,5], Seth Tjia[3], Armin Passecker[1,2], Katarzyna Buczak[2], Christin Gumpp[1,2], Alexander Schmidt[2], Matthias Rottmann[1,2], Christof Grüring[1,2], Matthias Marti[4,5], Zbynek Bozdech[3], Nicolas MB Brancucci[1,2]*

[1]Department of Medical Parasitology and Infection Biology, Swiss Tropical and Public Health Institute, Allschwil, Switzerland; [2]University of Basel, Basel, Switzerland; [3]School of Biological Sciences, Nanyang Technological University, Singapore, Singapore; [4]Wellcome Center for Integrative Parasitology, Institute of Infection, Immunity and Inflammation, University of Glasgow, Glasgow, United Kingdom; [5]Institute for Parasitology, University of Zurich, Zurich, Switzerland

*For correspondence:
nicolas.brancucci@swisstph.ch

**Abstract** *Plasmodium falciparum* accounts for the majority of over 600,000 malaria-associated deaths annually. Parasites resistant to nearly all antimalarials have emerged and the need for drugs with alternative modes of action is thus undoubted. The FK506-binding protein *Pf*FKBP35 has gained attention as a promising drug target due to its high affinity to the macrolide compound FK506 (tacrolimus). Whilst there is considerable interest in targeting *Pf*FKBP35 with small molecules, a genetic validation of this factor as a drug target is missing and its function in parasite biology remains elusive. Here, we show that limiting *Pf*FKBP35 levels are lethal to *P. falciparum* and result in a delayed death-like phenotype that is characterized by defective ribosome homeostasis and stalled protein synthesis. Our data furthermore suggest that FK506, unlike the action of this drug in model organisms, exerts its antiproliferative activity in a *Pf*FKBP35-independent manner and, using cellular thermal shift assays, we identify putative FK506-targets beyond *Pf*FKBP35. In addition to revealing first insights into the function of *Pf*FKBP35, our results show that FKBP-binding drugs can adopt non-canonical modes of action – with major implications for the development of FK506-derived molecules active against *Plasmodium* parasites and other eukaryotic pathogens.

## eLife assessment

FKBP35 is the only FK506-binding protein present in the malaria-causing parasite *Plasmodium falciparum*, and has been considered a promising drug target due to its high affinity to the macrolide compound FK506, an immunosuppressant with antiplasmodial activity. This study demonstrates the essentiality of FKBP35 in parasite growth, based on **compelling** genetic evidence. The data also suggest that FK506 may exert its antimalarial activity through FKBP35-independent mechanisms that have not yet been characterized. This **important** study will be of interest to scientists working on the parasite biology and antimalarial drug development.

## Introduction

Despite considerable progress in recent years, malaria remains one of the major global health threats (**WHO, 2022**). The apicomplexan parasite *Plasmodium falciparum* is responsible for the vast majority of the over 600,000 malaria deaths in 2020. Most of this burden is carried by infants and children under the age of five in sub-Saharan Africa. Malaria deaths increased by 12% compared to 2019,

which can at least in part be explained by the COVID-19-related suspension of malaria control and treatment measures (**WHO, 2022**). In addition, the emergence of parasites resistant to most antimalarials, including artemisinin-based combination therapies, endangers current and future disease elimination efforts and underscores the need for developing drugs with alternative modes of action (**Ippolito et al., 2021**).

FK506 (tacrolimus), a well-characterized immunosuppressant, shows considerable activity against asexual blood stage parasites (**Bell et al., 1994**). *P. falciparum* encodes a single FK506-binding protein (FKBP) dubbed *Pf*FKBP35 (PF3D7_1247400), which is considered to be a promising drug target (**Bharatham et al., 2011**; **Monaghan and Bell, 2005**). FKBPs belong to the immunophilin family and are conserved throughout the eukaryotic kingdom (**Bell et al., 2006**; **Kolos et al., 2018**). In contrast to the malaria parasite, most other eukaryotes encode several FKBPs that, besides showing high affinity for the FK506 drug, exert peptidyl-prolyl isomerase (PPIase) activity. The PPIase moiety catalyzes *cis-trans* isomerization of proline residues – an event thought to be rate-limiting for the folding of many proteins (**Kang et al., 2008**). Independent of their PPIase activity, FKBPs act as chaperones to prevent protein aggregation under stress conditions (**Bose et al., 1996**; **Furutani et al., 2000**). Furthermore, some FKBP variants play a role in cellular signaling or gene regulation (**Yang et al., 2001**; **Kasahara et al., 2020**). Human FKBP12, for instance, is best known for its ability to form a complex with the immunosuppressant drugs FK506 and rapamycin. These complexes inhibit the phosphatase activity of calcineurin and the kinase activity of mechanistic target of rapamycin (mTOR), respectively (**Cardenas et al., 1995**). Amongst many other effects, this inhibition results in decreased T-cell activation. Of note, the regulatory role of *Hs*FKBP12 depends on the presence of rapamycin or FK506. In the absence of drugs, this immunophilin neither interacts with mTOR nor with calcineurin and does hence not affect activity of these key components of eukaryotic cells. Due to the conserved nature of FKBPs, rapamycin and FK506 have also gained attention as potential antimicrobial drugs (**Bell et al., 1994**; **Kolos et al., 2018**).

Similar to its homologs in model organisms, *Pf*FKBP35 harbors PPIase and chaperoning activity in vitro (**Monaghan and Bell, 2005**). Despite sharing key features with FKBPs of other eukaryotes, its role in *P. falciparum* remains elusive. *Pf*FKBP35 is believed to interact with heat shock proteins (HSPs) and co-immunoprecipitates with cytoskeletal factors of *P. falciparum* (**Leneghan and Bell, 2015**). While *Pf*FKBP35 inhibits the phosphatase activity of recombinant calcineurin in an FK506-independent manner (**Yoon et al., 2007**; **Kumar et al., 2005**; **Alag et al., 2009**), it does not co-localize with calcineurin in vivo (**Kumar et al., 2005**). The protein is expressed throughout the 48 hr intra-erythrocytic development of *P. falciparum* and is found within the cytosol of ring stage parasites. In the older trophozoite and the replicating schizont stages, the protein was reported to also localize to the nucleus (**Kumar et al., 2005**). While a transposon-based mutagenesis screen suggests that *Pf*FKBP35 is essential for intra-erythrocytic replication (**Zhang et al., 2018**), neither its cellular function nor its essentiality was confirmed experimentally in live parasites. The unknown role in parasite biology notwithstanding, *Pf*FKBP35 is considered to be a viable drug target and, given the fact that *P. vivax* and *P. knowlesi* encode FKBP homologs that exhibit comparable PPIase activities (**Alag et al., 2010**; **Goh et al., 2018**), future FKBP-targeted therapies may be effective across different *Plasmodium* species.

Crystal structures of *Pf*FKBP35 in complex with FK506 and rapamycin revealed high-affinity interactions with the FK506-binding domain of the protein (**Yoon et al., 2007**; **Bianchin et al., 2015**; **Kotaka et al., 2008**). Consistent with these observations, in vitro enzyme activity assays showed that FK506 inhibits the PPIase activity of recombinant *Pf*FKBP35 in a dose-dependent manner (**Monaghan and Bell, 2005**). Previous research aimed at exploiting structural differences between human and parasite-encoded FKBPs to circumvent the immunosuppressive activity of drug-bound *Hs*FKBP12 (**MacDonald and Boyd, 2015a**; **Harikishore et al., 2013b**; **Harikishore et al., 2013a**; **Rajan and Yoon, 2022**). These efforts include investigation of non-covalent FKBP inhibitors such as adamantyl derivatives, macrocycles, the small molecule D44 (**MacDonald and Boyd, 2015a**; **Harikishore et al., 2013b**; **Harikishore et al., 2013a**; **MacDonald and Boyd, 2015b**; **Deepa and Thirumeignanam, 2020**) and the synthetic ligand for FKBP (SLF) as well as derivatives thereof designed to covalently bind *Pf*FKBP35 (**Atack et al., 2020**). While D44 shows promising antimalarial activity, it fails to alter thermostability of the FK506-binding domains of *Pf*FKBP35 and *Hs*FKBP12 in vitro (**Atack et al., 2020**), suggesting that this compound does not directly interact with *Pf*FKBP35. Up until now, *Pf*FKBP35 has not been

validated as a drug target using reverse genetics, and the link between *Pf*FKBP35-interacting drugs and their antimalarial activity remains elusive (*Bell et al., 1994*).

Here, we generated inducible *Pf*FKBP35 knock-out, knock-down, as well as overexpression cell lines and demonstrate that *Pf*FKBP35 is essential for asexual replication of blood forms. We show that the knock-out of *Pf*FKBP35 results in reduced levels of ribosomal proteins and stalled protein synthesis. However, these effects are not reflected in the parasite transcriptome, indicating a vital role at the post-transcriptional level. While the mechanistic link between *Pf*FKBP35 and ribosome homeostasis remains elusive, our data underpins the potential of *Pf*FKBP35 as a drug target and, using cellular thermal shift assays (CETSA), we corroborate the high-affinity binding of FK506 to *Pf*FKBP35. Importantly, however, our data does not support scenarios in which FK506 inhibits the essential function of *Pf*FKBP35, suggesting that it exerts its antimalarial activity in a *Pf*FKBP35-independent manner, which has major implications for the development of future FK506-based antimalarials and/or *Pf*FKBP35-inhibiting drugs.

## Results

### *Pf*FKBP35 is essential for asexual replication

In a first step, using CRISPR/Cas9-mediated gene editing, we engineered a *P. falciparum* cell line allowing for a conditional knock-down of *Pf*FKBP35 using the DD/Shield system (*Armstrong and Goldberg, 2007*; *Banaszynski et al., 2006*; *Figure 1A*, *Figure 1—figure supplement 1*). Despite the efficient depletion of *Pf*FKBP35 under knock-down conditions, parasites did not show apparent growth defects (*Figure 1—figure supplement 1*). Immunofluorescence assays (IFA) showed that *Pf*FKBP35 localizes to distinct foci within the parasite nucleus (*Figure 1B*).

Next, we generated a cell line allowing for the conditional knock-out of the endogenous coding sequence using a dimerizable Cre recombinase (DiCre)-based system (*Collins et al., 2013*). Specifically, we introduced two *loxP* sites – one within a synthetic intron (*Jones et al., 2016*) downstream of the start codon and one after the stop codon of the *fkbp35* coding region – and added a *green fluorescent protein* (*gfp*) sequence downstream of this expression cassette (*Figure 1C*, *Figure 1—figure supplement 2*). As a result, these NF54/iKO-FKBP35 parasites express tag-free, wild-type (WT) *Pf*FKBP35 protein under control conditions (FKBP35$^{WT}$). Upon DiCre activation, the *loxP* sites are recombined and the *fkbp35* locus is excised from the parasite genome (*Figure 1C*, *Figure 1—figure supplement 2*). Under these knock-out conditions (FKBP35$^{KO}$), parasites express cytosolic GFP under control of the endogenous *fkbp35* promoter. This setup allows monitoring knock-out efficiency at the single cell level and revealed that DiCre successfully deleted *fkbp35* in the vast majority of parasites (*Figure 1D*).

In a first attempt, we induced the *fkbp35* knock-out in synchronous ring stage NF54/iKO-FKBP35 parasites at 0–6 hr post erythrocyte invasion (hpi), that is, prior to the expected peak transcription of *fkbp35* (*Kucharski et al., 2020*). These FKBP35$^{KO}$ parasites showed a delayed death-like phenotype. In the first generation (G1), parasites completed intra-erythrocytic development without noticeable effects (*Figure 1E*) and entered the next generation (G2) with a multiplication rate similar to that of the FKBP35$^{WT}$ control population (*Figure 1F*). In G2, however, FKBP35$^{KO}$ parasites arrested at the late trophozoite/early schizont stage (approximately 30–36 hpi) with many cells failing to enter DNA replication in the S-phase (*Figure 1E*). To investigate this phenomenon in more detail, we induced the *fkbp35* knock-out at three different time points during the IDC. Similar to DiCre activation at 0–6 hpi, knocking out *fkbp35* at 12–18 hpi resulted in parasite death at the trophozoite/schizont stage of the subsequent generation. However, when induced at 24–30 hpi or 34–40 hpi in G1, parasites did not only complete the current IDC but also successfully passed through G2 before they eventually arrested in G3 (*Figure 1G*). Considering that low *Pf*FKBP35 levels are sufficient to maintain the essential function of the protein, as observed with the NF54/iKD-FKBP35 parasites (see *Figure 1—figure supplement 1*), it is conceivable that proteins expressed between 0 and 24 hpi, i.e. prior to deleting *fkbp35* from the genome, allow parasites to enter G3.

The delayed death of FKBP35$^{KO}$ is reminiscent of a phenotype observed in apicoplast-deficient parasites (*Wiley et al., 2015*; *Kennedy et al., 2019*). We therefore tested if *Pf*FKBP35 is involved in apicoplast biology, that is, the synthesis and/or export of isoprenoid precursors, but did not find evidence for an association with this organelle (*Figure 1—figure supplement 2*). Furthermore, given

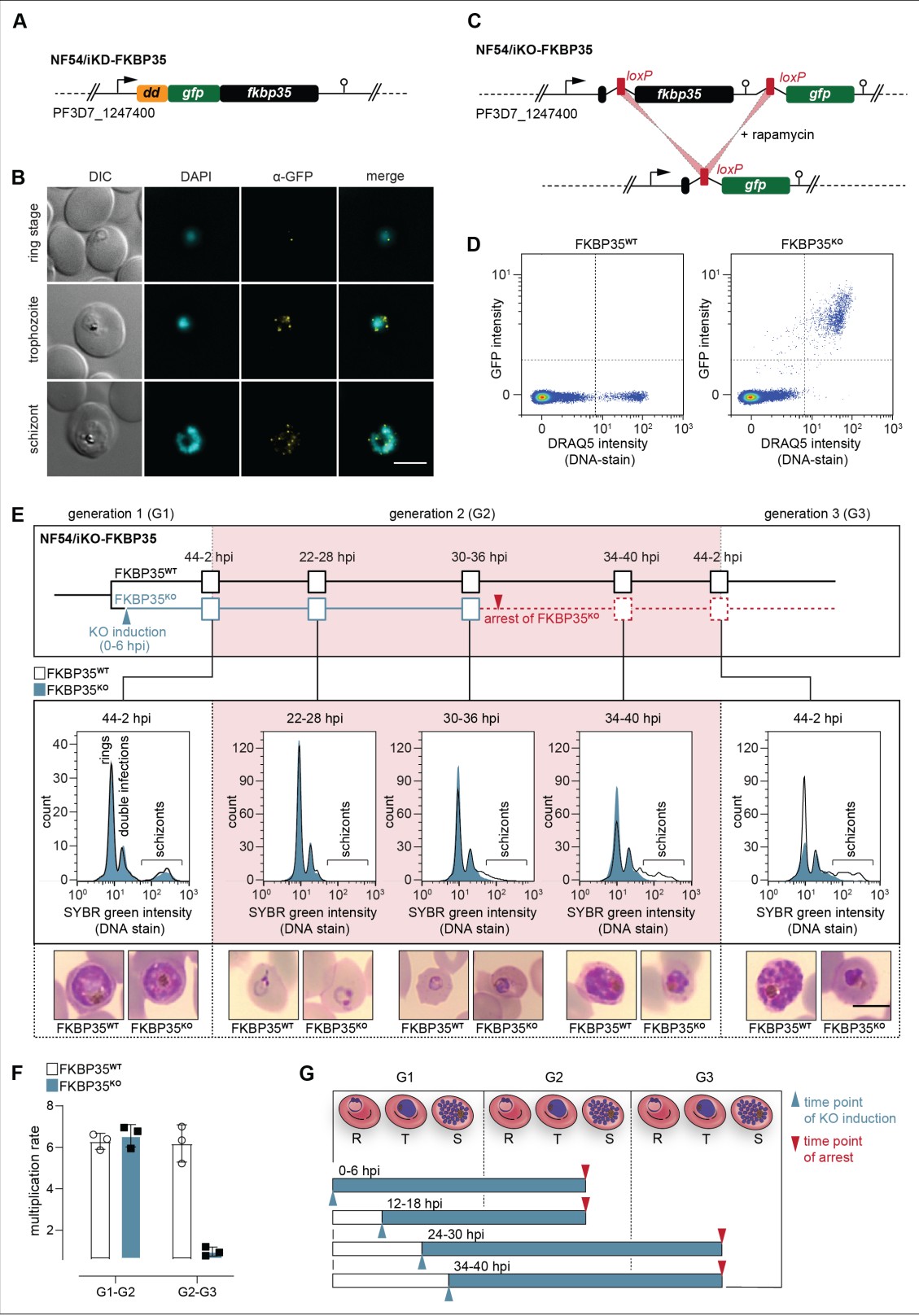

**Figure 1.** *Pf*FKBP35 depletion causes delayed parasite death. (**A**) Schematic of the N-terminally modified *fkbp35* locus in NF54/iKD-FKBP35 parasites. *dd*, destabilization domain; *gfp*, green fluorescent protein. (**B**) Localization of *Pf*FKBP35 in NF54/iKD-FKBP35 parasites at the ring, trophozoite, and schizont stage assessed by immunofluorescence. DNA was stained with DAPI. Representative images are shown. Scale bar: 5 µm. DIC, differential interference contrast. (**C**) Schematic of the modified *fkbp35* locus in NF54/iKO-FKBP35 parasites before and after *loxP* recombination. *gfp*, *green*

*Figure 1 continued on next page*

*Figure 1 continued*

*fluorescent protein.* (**D**) Flow cytometry plot showing GFP expression in G1 schizonts of FKBP35$^{KO}$ and FKBP35$^{WT}$ parasites. DNA was stained with DRAQ5. (**E**) Time course showing the development of NF54/iKO-FKBP35 parasites. Upper panel: schematic of the experimental setup. Middle panel: DNA content of the populations assessed by flow cytometry based on SYBR Green intensity. Lower panel: Giemsa-stained blood smears. The knock-out was induced at 0–6 hpi. Representative data of three biological replicates are shown. Scale bar: 5 µm. (**F**) Multiplication rates of FKBP35$^{KO}$ and FKBP35$^{WT}$ parasites from generation 1 to generation 2 (G1–G2) and G2–G3. The knock-out was induced at 0–6 hpi. Multiplication rates were determined using flow cytometry. Data points represent the multiplication rates from G1–G2 and G2–G3. n = 3, error bars represent the standard deviation. (**G**) Schematic showing time point of growth arrest (red arrow) following *Pf*FKBP35 knock-out at different hpi (blue arrow). R, ring stage; T, trophozoite stage; S, schizont stage.

The online version of this article includes the following source data and figure supplement(s) for figure 1:

**Figure supplement 1.** Characterization of NF54/iKD-FKBP35 parasites.

**Figure supplement 1—source data 1.** Agarose gel electrophoresis of peqGREEN-stained DNA and SDS-polyacrylamide gel electrophoresis (SDS-PAGE) of protein samples.

**Figure supplement 2.** CRISPR/Cas9-based generation of transgenic parasite lines.

**Figure supplement 2—source data 1.** Agarose gel electrophoresis of peqGREEN-stained DNA.

**Figure supplement 3.** Gating strategy of flow cytometry data.

the reported interaction of *Pf*FKBP35 with heat shock proteins (*Leneghan and Bell, 2015*; *Kumar et al., 2005*), we also tested the parasite`s response to heat stress, but could not detect altered heat-susceptibility between FKBP35$^{KO}$ and FKBP35$^{WT}$ cells (*Figure 1—figure supplement 2*).

## *Pf*FKBP35 is crucial for ribosome homeostasis and protein synthesis

To further elucidate the function of *Pf*FKBP35, we induced the *fkbp35* knock-out in early ring stages (0–6 hpi) and compared the proteomes of FKBP35$^{KO}$ and FKBP35$^{WT}$ parasites at the schizont stage of G1 (36–42 hpi) and at the trophozoite stage of the following generation (G2, 24–30 hpi) using a quantitative proteomics approach.

### The knock-out of *Pf*FKBP35 causes accumulation of pre-ribosome components in G1

Consistent with the efficient deletion of *fkbp35* on the genomic level, FKBP35$^{KO}$ schizonts of G1 showed a 19-fold (18.9 ± 5.7 s.d.) reduction of FKBP35 and GFP levels were increased by 84-fold (83.7 ± 38.5 s.d.) when compared to the control population (*Figure 2A*, *Supplementary file 1*). Besides *Pf*FKBP35 and GFP, 50 parasite proteins were significantly deregulated in FKBP35$^{KO}$ schizonts and 34 of them were found at elevated levels compared to the FKBP35$^{WT}$ control conditions. A gene ontology analysis using PANTHER (*Mi et al., 2021*) indicated enrichment of the '*pre-ribosome*' (fold enrichment = 30.4, p-value=1.46E-5, 4/34 proteins) and '*nucleolus*' (fold enrichment = 6.77, p-value=8.85E-4, 5/34 proteins) terms among these factors. While the pre-ribosome is a large subunit precursor of mature ribosomes, the nucleolus represents the primary site for the biogenesis of ribosomal subunits (*Boisvert et al., 2007*). In contrast to factors being more abundant in knock-out parasites, none of the 16 factors with lower abundance in FKBP35$^{KO}$ cells were associated with ribosome-related processes (*Supplementary file 1*). Despite being phenotypically silent in G1, the accumulation of pre-ribosome and nucleolar components hints at perturbed ribosome maturation in these cells. Specifically, it is conceivable that the observed buildup of ribosome intermediates results from reduced anabolic activity, that is, the stalling of nucleolar pre-ribosome assembly (*Nissan et al., 2002*).

Notwithstanding these effects, FKBP35$^{KO}$ cells were equally sensitive to the translation inhibitor cycloheximide compared to their FKBP35$^{WT}$ counterparts (*Figure 2B*), indicating that ribosome function is not compromised at this point in time (*Plouffe et al., 2008*). This is consistent with normal cell cycle progression in G1 and unaltered multiplication rates of FKBP35$^{KO}$ parasites, as described above (see *Figure 1E and F*), and shows that the prominent depletion of *Pf*FKBP35 does not have a major impact on parasite viability in G1.

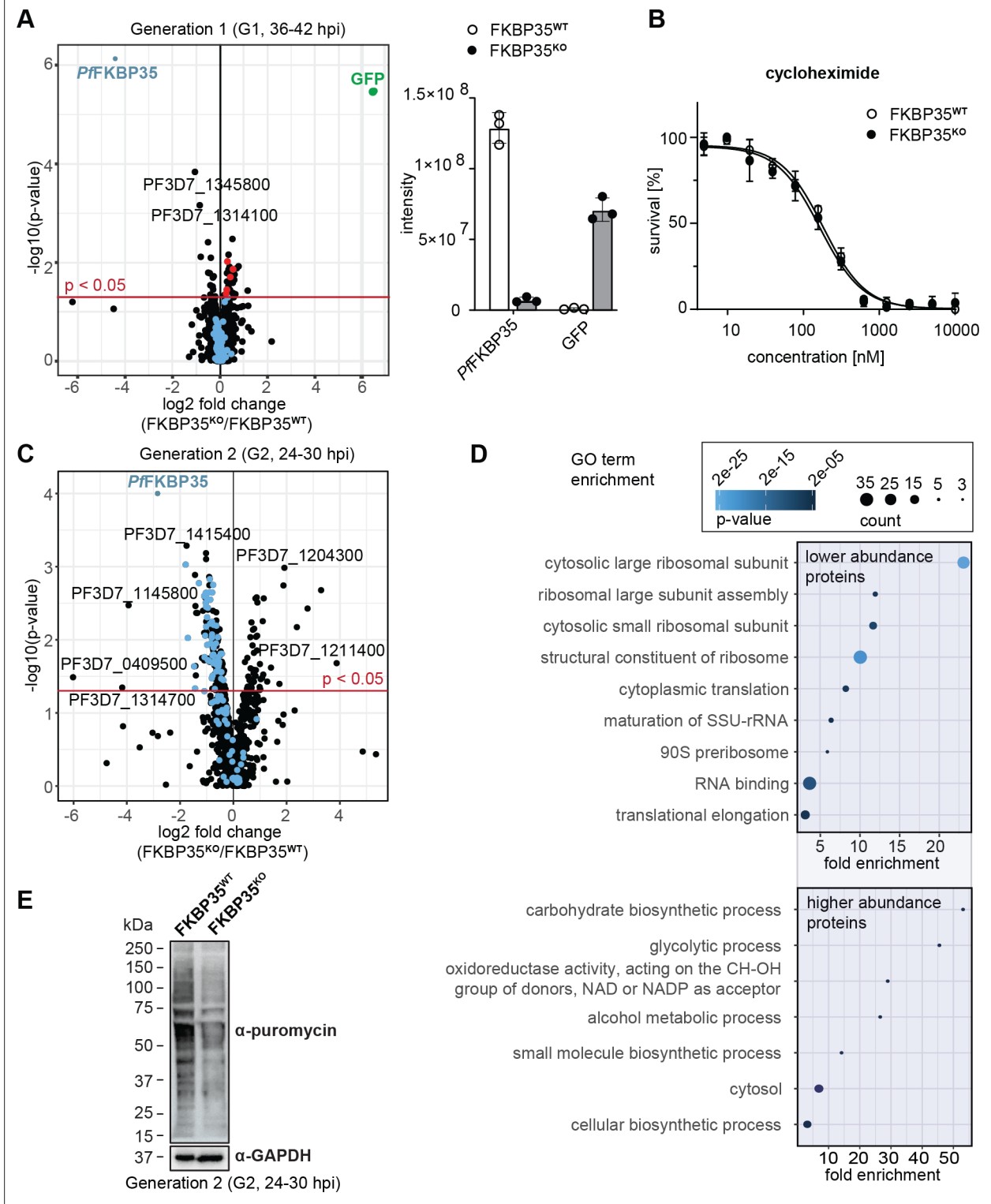

**Figure 2.** *Pf*FKBP35 is crucial for ribosome homeostasis and protein synthesis. (**A**) Left panel: Proteome analysis of schizonts (36–42 hpi) in G1. The volcano plot shows change in protein abundance in FKBP35^KO compared to FKBP35^WT. Ribosomal proteins are highlighted in blue, proteins associated with the GO terms 'pre-ribosome' or 'nucleolus' in red. *P*-values were calculated using empirical Bayes moderated t-tests and adjusted for multiple testing using the Benjamini-Hochberg method. n = 3. Right panel: relative difference in *Pf*FKBP35 and GFP abundance expressed as signal intensity measured by mass spectrometry. n = 3, error bars represent the standard deviation. (**B**) Dose–response effect of the translation inhibitor cycloheximide on FKBP35^KO and FKBP35^WT parasites. The knock-out was induced at 0–6 hpi and parasites were subsequently exposed to cycloheximide for 48 hr.

*Figure 2 continued on next page*

*Figure 2 continued*

Parasite survival was quantified by flow cytometry. n = 3, error bars represent the standard error of the mean. (**C**) Proteome analysis of trophozoites (24–30 hpi) in G2. The volcano plot shows change in protein abundance in FKBP35**KO** compared to FKBP35**WT**. Ribosomal proteins are highlighted in blue. *P*-values were calculated using empirical Bayes moderated t-tests and adjusted for multiple testing using the Benjamini-Hochberg method. n = 3. (**D**) GO enrichment analysis of significantly deregulated proteins in FKBP35**KO** trophozoites in G2. GO enrichment analysis was performed using the PANTHER 'Overrepresentation Test' with the test type 'Fisher's exact' and the correction method 'false discovery rate.' (**E**) Assessment of translation activity using SUnSET. FKBP35**KO** and FKBP35**WT** trophozoites of G2 were incubated with puromycin for 1 hr. Incorporated puromycin was detected using an α-puromycin antibody. α-GAPDH served as loading control. n = 3, a representative Western blot is shown.

The online version of this article includes the following source data and figure supplement(s) for figure 2:

**Source data 1.** SDS-polyacrylamide gel electrophoresis (SDS-PAGE) of protein samples.

**Figure supplement 1.** Assessment of translation rates using SUnSET.

**Figure supplement 1—source data 1.** SDS-polyacrylamide gel electrophoresis (SDS-PAGE) of protein samples.

## *Pf*FKBP35 is essential for maintaining normal levels of functional ribosomes in G2

In the following generation (G2, 24–30 hpi), the proteome of FKBP35**KO** trophozoites showed substantial differences. Despite the absence of morphological changes at this point in time, 216 proteins were significantly deregulated (*Figure 2C*). Among these, 50 proteins were found at higher abundance compared to FKBP35**WT** and their gene ontology terms point toward an effect on processes involved in energy metabolism (*Figure 2D*).

The majority of deregulated factors (166 of 216) in G2, however, were less abundant in FKBP35**KO** compared to the FKBP35**WT** control cells. Very prominently, 20 of the top 50 hit molecules are ribosomal proteins that, on average, show a twofold reduction (2.0 ± 0.4 s.d.) in FKBP35**KO** parasites (*Supplementary file 1*). It is therefore not surprising that PANTHER identified the knock-out of *Pf*FKBP35 to mainly affect ribosome-related processes, such as '*cytoplasmic translation*,' '*ribosome assembly*,' and '*ribosomal large subunit biogenesis*' (*Figure 2D*; *Mi et al., 2021*).

To assess if the reduced levels of ribosomal proteins result in lower protein synthesis rates, we employed the 'surface sensing of translation' (SUnSET) technique. SUnSET is based on incorporation of the tyrosyl-tRNA analogue puromycin into nascent peptide chains and thereby allows monitoring newly synthesized proteins (*Schmidt et al., 2009*). As expected, we found that FKBP35**KO** trophozoites in G2 (24–30 hpi) showed reduced translation rates compared to the FKBP35**WT** control population (*Figure 2E*), while protein synthesis was not affected under control conditions (rapamycin-induced DiCre activation in the NF54/DiCre parental line (*Figure 2—figure supplement 1*)).

Taken together, these results indicate that, under knock-out conditions, *Pf*FKBP35 levels become limiting for ribosome maturation late during schizogony of G1, without having immediate effects on the steady-state levels of mature ribosomes. The knock-out of *Pf*FKBP35 is thus phenotypically silent in G1. In the absence of a continuous supply of mature ribosomes under FKBP35**KO** conditions, however, FKBP35**KO** parasites fail at maintaining translation at a level that would support normal parasite development in the next generation (G2).

## The knock-out of *Pf*FKBP35 is transcriptionally silent

To examine if the altered proteome of FKBP35**KO** parasites is the result of *Pf*FKBP35-controlled transcription, we compared the transcriptional profiles of FKBP35**WT** and FKBP35**KO** in a time-course RNA-sequencing (RNA-Seq) experiment spanning two IDCs. Specifically, we induced the knock-out of FKBP35 in synchronous parasites at 0–6 hpi and collected RNA from parasite populations at three and four subsequent time points (TPs) in G1 and G2, respectively (*Figure 3A*, *Figure 3—figure supplement 1*). With the exception of TP7, these time points cover a time frame during which FKBP35**KO** cells cannot be distinguished from their FKBP35-expressing counterparts on a morphological level (compare *Figure 1E*).

As expected, the transcriptome of FKBP35**KO** showed a prominent reduction of steady-state *fkbp35* transcripts already in TP1 (11.3-fold ± 2.0 s.d.) compared to the FKBP35**WT** control population (*Figure 3—figure supplement 1*, *Supplementary file 2*). During peak transcription at 30–36 hpi (TP2), this difference increased to 240-fold (238.4 ± 20.0 s.d.) and, consistent with the efficient recombination at the *fkbp35* locus, these cells started transcribing *gfp*. Notwithstanding the prominent

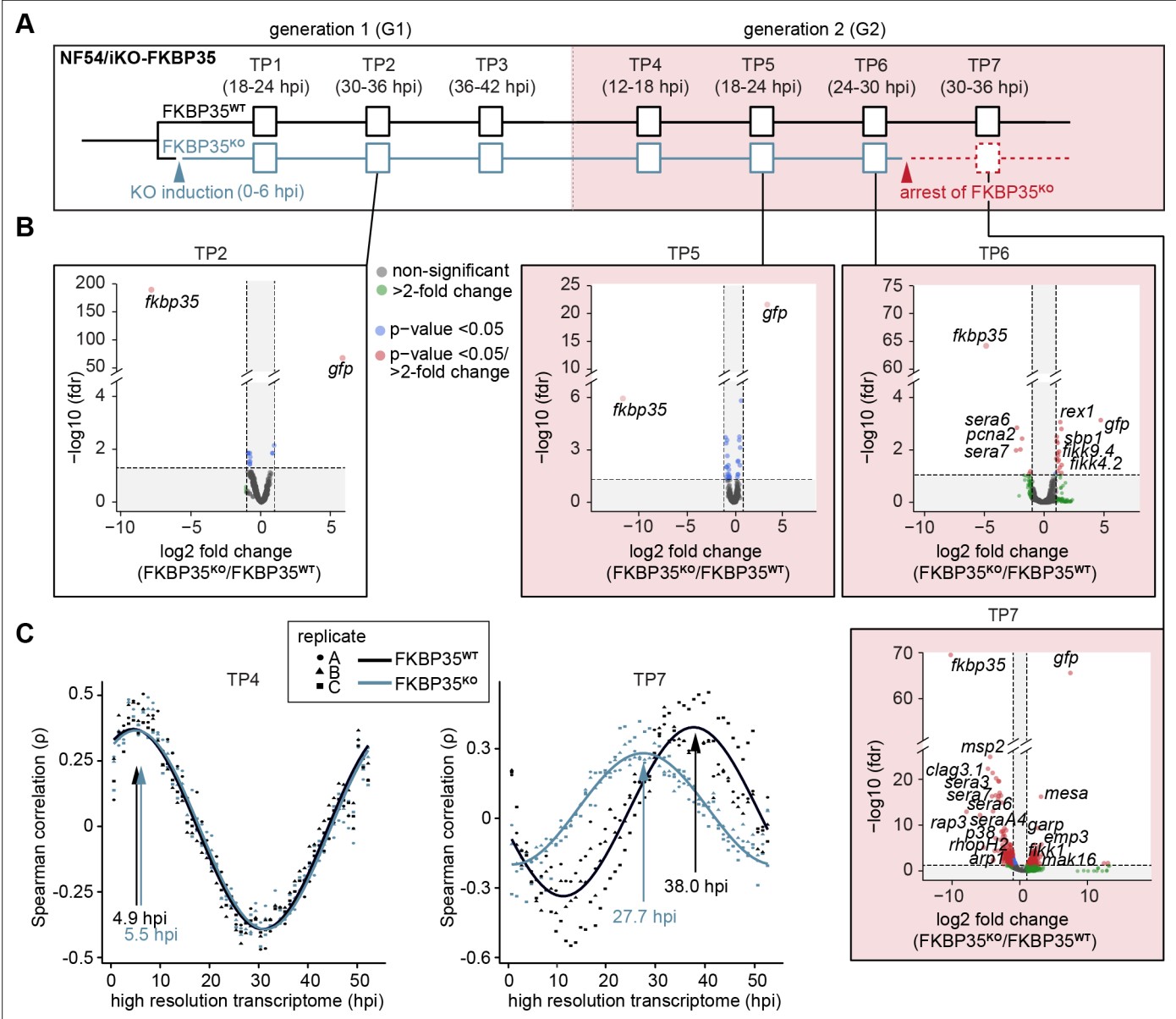

**Figure 3.** The knock-out of *Pf*FKBP35 remains silent at the transcriptomic level. (**A**) Setup of the RNA-seq experiment. RNA was collected at seven indicated time points (TP). (**B**) Volcano plots show differential gene expression between FKBP35^KO and FKBP35^WT parasites. Genes are color-coded according to the fold change and false discovery rate (FDR). Paired differential expression tests were performed using the DESeq2 likelihood ratio test. n = 3. (**C**) Mapping of FKBP35^KO and FKBP35^WT transcriptomes to a high-resolution reference (*Bozdech et al., 2003*). Spearman rank coefficients ( $\rho$ ) show correlation between the sampled transcriptomes and the reference dataset (*Bozdech et al., 2003*). The TPs with the highest correlation coefficient are indicated.

The online version of this article includes the following figure supplement(s) for figure 3:

**Figure supplement 1.** Transcriptomics reveals a cell cycle arrest of FKBP35^KO parasites.

depletion of *fkbp35* transcripts, the transcriptome of FKBP35^KO is only marginally affected up to and including 24–30 hpi in G2 (TP6). By contrast, at 30–36 hpi (TP7) the transcription of knock-out parasites shows major changes (*Figure 3B*). Considering the appearance of morphological changes in FKBP35^KO at this time point, effects on the transcriptome are unlikely to represent specific activities of *Pf*FKBP35 in gene regulation. Indeed, mapping the transcriptome of individual time points to a high-resolution reference (*Bozdech et al., 2003*) revealed a significant slowdown of FKBP35^KO compared to FKBP35^WT parasites at TP7 (*Figure 3C*, *Figure 3—figure supplement 1*). This indicates aborted cell cycle progression in FKBP35^KO parasites during a phase that is characterized by a rapid increase

in biomass – possibly due to a translation and/or size-dependent cell cycle checkpoint (*Barnum and O'Connell, 2014*). At earlier time points (TP1–TP6), the transcriptomes of FKBP35[KO] and FKBP35[WT] are highly comparable and the differences observed on the level of individual genes clearly fail at explaining the proteomic changes observed in FKBP35[KO] parasites at 24–30 hpi in G2 (TP6), reiterating that the essential activity of *Pf*FKBP35 is linked to transcription-independent processes.

## Probing FK506 interactions in cellular thermal shift assays (CETSA)

*Pf*FKBP35 emerged as a promising antimalarial drug target (*Bharatham et al., 2011*) and considerable efforts were made to define interactions of this immunophilin with small molecules in the last decade (*Harikishore et al., 2013b*; *Harikishore et al., 2013a*; *Rajan and Yoon, 2022*; *Monaghan et al., 2017*). Several studies conclusively demonstrated that FK506 – the most well-known ligand of mammalian FKBPs (*Van Duyne et al., 1991*) – binds to *Pf*FKBP35 (*Yoon et al., 2007*; *Bianchin et al., 2015*; *Kotaka et al., 2008*) and kills asexual blood stage parasites with a half-maximal inhibitory concentration (IC50) in the low micromolar range at 1.9 µM (*Bell et al., 1994*).

To investigate interactions of FK506 with *P. falciparum* proteins, we used the *cellular thermal shift assay followed by mass spectrometry* (MS-CETSA). This approach exploits the fact that, once drug–protein complexes are formed, thermostability of the target protein is altered (*Dziekan et al., 2019*). We applied the so-called *isothermal dose–response* variant of CETSA using FK506 (*Dziekan et al., 2020*). Here, target proteins are identified based on their dose-dependent stabilization by drugs under thermal challenge (different temperatures) when compared to a non-denaturing condition (37°C) using the change in the area under the curve (ΔAUC) and the dose–response curve goodness of fit ($R^2$) as metrics (*Figure 4A*; *Dziekan et al., 2020*).

### CETSA identifies targets of FK506

In a first step, we performed CETSA in combination with drug-exposed protein extracts. Since proteins lose their physiological context during the process of protein extraction, this variant of the CETSA approach is designed to identify direct protein–drug interactions (*Dziekan et al., 2019*; *Dziekan et al., 2020*). Not surprisingly, we found *Pf*FKBP35 to be stabilized by FK506 in a dose-dependent manner when challenged at 60°C with a half maximal effective concentration (EC50) of 152.0 nM (*Figure 4B*). The fact that FK506 exhibits its stabilizing activity already in the low nanomolar concentration range indicates high-affinity interactions between the protein–drug pair.

Besides *Pf*FKBP35, FK506 stabilizes a number of other proteins in a dose-dependent manner (*Figure 4C* and *Figure 4—figure supplement 1*). A few of these factors show pronounced sigmoidal stabilization profiles and are predicted to be essential (*Figure 4C*; *Zhang et al., 2018*). Amongst others, N6-methyl transferase MT-A70 (PF3D7_0729500), histone deacetylase HDA2 (PF3D7_1008000), pre-mRNA-splicing factor CEF1 (PF3D7_1033600), FeS cluster assembly protein SufD (PF3D7_1103400), and the H/ACA ribonucleoprotein complex subunit 1 (PF3D7_1309500) showed most pronounced stabilization profiles (*Figure 4—figure supplement 1*, *Supplementary file 3*). In contrast to *Pf*FKBP35, most of these likely essential candidates are stabilized at FK506 concentrations close to the drug's IC50 (*Bell et al., 1994*). It can therefore not be excluded that the binding of FK506 to at least one of these factors contributes to the antimalarial activity of the drug.

### The *Pf*FKBP35/FK506 pair interferes with ribosomal complexes

Given the high-affinity interaction of *Pf*FKBP35 with FK506 and its association with ribosomal proteins, we set out to describe the native context of *Pf*FKBP35 using the 'live cell variant' of CETSA. In contrast to focusing exclusively on direct protein–drug interactions, live cell CETSA allows capturing drug-induced shifts in protein stability beyond those of direct binding partners (*Dziekan et al., 2020*) and amongst others allows identifying factors present in the same protein complex as the target protein. To do so, CETSA takes advantage of the fact that factors within complexes have similar denaturation temperatures due to their co-aggregation upon heat-induced denaturation (*Tan et al., 2018*). As a consequence, this CETSA variant generally identifies more factors compared to the protein extract-based CETSA approach (*Figure 4A*).

As expected, and confirming the results above, live cell CETSA shows that FK506 alters the thermal stability of *Pf*FKBP35 already at very low drug concentrations when challenged at 60°C with an EC50 of 286.7 nM (*Figure 4B*). Interestingly, and in contrast to the protein lysate-based approach, live cell

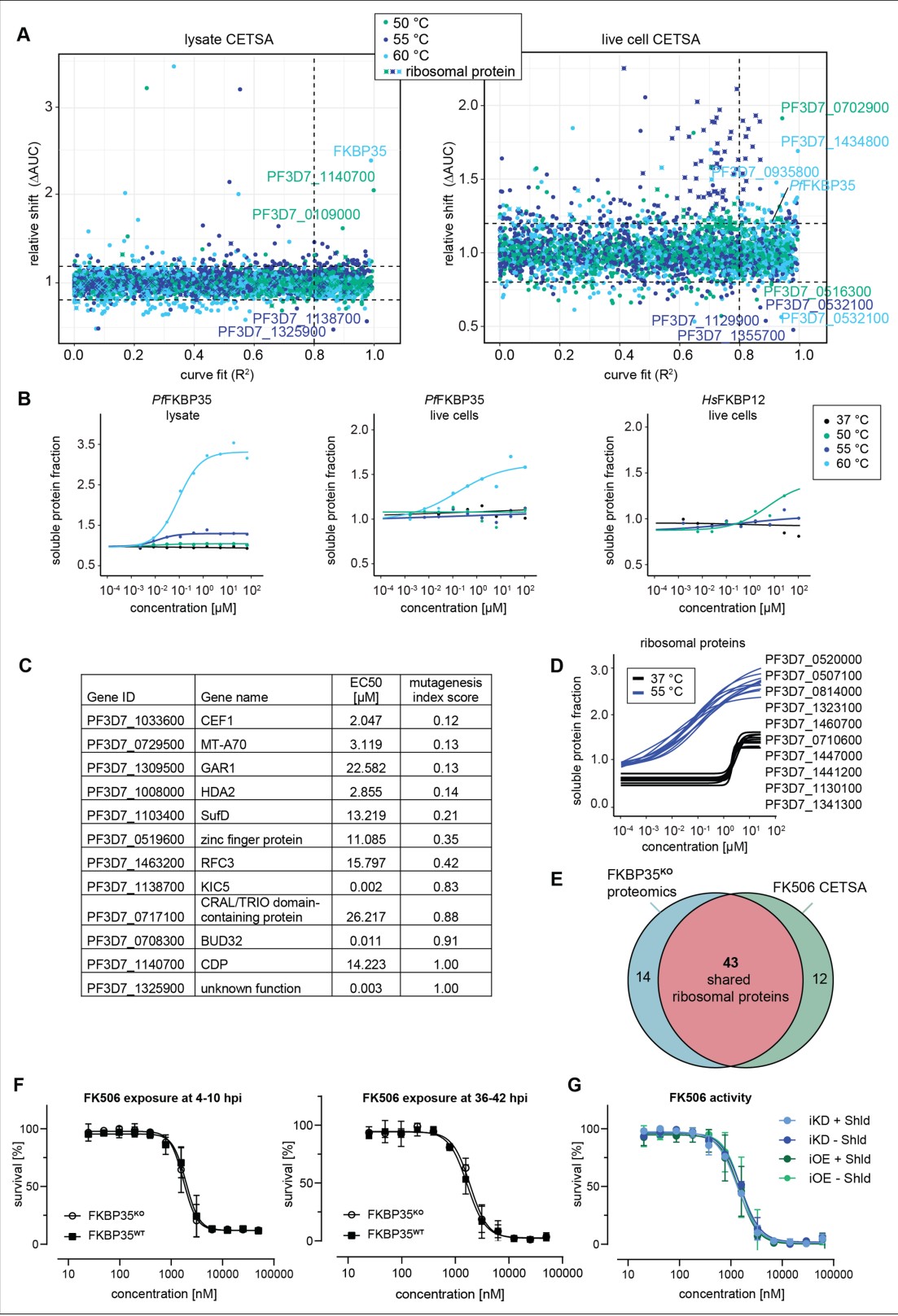

**Figure 4.** Cellular thermal shift assays (CETSA) identify FK506 interaction partners. (**A**) CETSA results. The relative shift in protein abundance (area under the curve of heat-challenged samples normalized against the non-denaturing control, ΔAUC) as a function of $R^2$ (goodness of fit as a measure for the dose–response effect). The dashed lines indicate an arbitrary cutoff of three times the median absolute deviation (MAD) of each dataset and $R^2 = 0.8$. Three different temperatures were used for protein denaturation (indicated by colors). (**B**) Dose-dependent stabilization of *P. falciparum*

*Figure 4 continued on next page*

*Figure 4 continued*

and human FKBPs detected using lysate and live cell CETSA. Bullets represent the abundance of non-denatured proteins in response to increasing FK506 concentrations under thermal challenge relative to the DMSO control. Colors indicate the temperatures used for heat challenge; the black curve represents the non-denaturing (37°C) control. (**C**) Proteins with the most pronounced stability shift under thermal challenge in the lysate CETSA experiment are listed. The EC50 is the concentration at which the half-maximal effect of the FK506-induced thermal stabilization is reached. The mutagenesis index score determined by *Zhang et al., 2018* is a measure to predict if a protein is essential for asexual growth: 0 = essential, 1 = non-essential. (**D**) Fitted live cell CETSA curves of 10 representative ribosomal proteins at 37°C and 55°C. Ribosomal proteins show an FK506 dose-dependent increase in the soluble protein fraction. (**E**) Venn diagram showing the overlap between ribosomal proteins deregulated in the *Pf*FKBP35 knock-out cell line (less abundant in G2 FKBP[KO] trophozoites determined by proteomics) and the ribosomal proteins stabilized by FK506 in live cell CETSA. The number of factors identified in either or both assays is indicated. (**F**) Dose–response effect of FK506 on FKBP35[KO] and FKBP35[WT] parasites. The knock-out was induced at 0–6 hpi and parasites were exposed to FK506 at 4–10 hpi (left panel) and 36–42 hpi in G1 (right panel). Parasite survival was measured after reinvasion (10–16 hpi) in G2. n = 3, error bars represent the standard error of the mean. (**G**) Dose–response effect of FK506 on NF54/iKD-FKBP35 and NF54/iOE-FKBP35 parasites in the presence and absence of Shield-1 (Shld). Parasites were split at 0–6 hpi and subsequently treated with Shld or the vehicle control EtOH. Data points represent the mean of three and two independent biological replicates for the NF54/iOE-FKBP35 and NF54/iKD-FKBP35 cell line, respectively. Error bars represent the standard error of the mean.

The online version of this article includes the following source data and figure supplement(s) for figure 4:

**Figure supplement 1.** Putative FK506 interaction partners identified by protein lysate-based cellular thermal shift assays (CETSA).

**Figure supplement 2.** Putative FK506 interaction partners identified by live cell cellular thermal shift assays (CETSA).

**Figure supplement 3.** Characterization of NF54/iOE-FKBP35 parasites.

**Figure supplement 3—source data 1.** Agarose gel electrophoresis of peqGREEN-stained DNA and SDS-polyacrylamide gel electrophoresis (SDS-PAGE) of protein samples.

**Figure supplement 4.** FKBP-targeting drugs fail at inhibiting parasite *Pf*FKBP35.

**Figure supplement 5.** *Pf*FKBP35 localizes near fibrillarin-dense nuclear regions.

CETSA revealed that more than 50 ribosomal proteins are stabilized in the presence of FK506 when challenged at 55°C, with a median EC50 of 45.9 nM (25th percentile = 34.2, 75th percentile = 58.9). Of note, these ribosomal proteins were stabilized at virtually identical FK506 concentrations (*Figure 4D*, *Figure 4—figure supplement 2*), indicating that the drug – directly or indirectly – interacts with ribosomal complexes. This is in further agreement with the fact that these ribosomal factors were not identified by protein lysate-based CETSA (compare *Figure 4A* and *Figure 4—figure supplement 1*).

Worth mentioning, we observed a drug dose-dependent increase in soluble protein abundance of ribosomal proteins also under non-denaturing conditions, that is, at 37°C, with a median EC50 of 1.94 µM (25th percentile = 1.85, 75th percentile = 2.1) (*Figure 4D*, *Figure 4—figure supplement 2*). Generally, due to the absence of a denaturing heat challenge, such a drug-dependent enrichment in soluble protein levels does not indicate binding-induced thermal stabilization of proteins (*Dziekan et al., 2020*). Instead, this can be the result of increased protein synthesis, altered protein conformation or, alternatively, occur in response to factors disengaging from larger protein complexes. The fact that FK506 solubilizes a high number of ribosomal proteins within a very narrow concentration window also at 37°C implies that these factors are liberated from the same ribosomal complex or from complexes that interact with each other and shows that FK506 negatively affects stability of ribosomal structures in live cells. However, these effects are limited to high, presumably saturating FK506 concentrations. Importantly, the vast majority of ribosomal proteins identified in live cell CETSA are shared with those de-regulated in *Pf*FKBP35 knock-out cells (*Figure 4E*), pointing toward a shared localization of FK506 and its *Pf*FKBP35 target in live parasites.

Together with the above-described importance of *Pf*FKBP35 in ribosome homeostasis, these results strongly suggest that FK506 is recruited to the activity site of *Pf*FKBP35 – most probably the nucleolar maturation sites of ribosomes – where it interacts with ribosomal complexes.

## FKBP-targeting drugs act in an *Pf*FKBP35-independent manner

Considering the proposed direct interaction between FK506 and FKBP35 in *P. falciparum*, we expected that FKBP35[KO] parasites show altered sensitivity to the drug. Consistent with published data, FK506 killed FKBP35[WT] cells with an IC50 of 1942 nM (+/-193 nM s.d.) when administered at 4–10 hpi in G1 (*Figure 4F*). Surprisingly however, and despite the prominent depletion of the protein under knock-out conditions, FKBP35[KO] and FKBP35[WT] parasites were equally sensitive to FK506 (IC50 = 1741 nM ± 187 s.d.; *Figure 4F*; *Bell et al., 1994*). Similarly, neither the efficient knock-down of

*Pf*FKBP35 in NF54/iKD-FKBP35 parasites (compare *Figure 1—figure supplement 1*) nor its overexpression (OE) in NF54/iOE-FKBP35 cells (*Figure 4—figure supplement 3*) resulted in the expected sensitivity change to FK506 treatment (*Figure 4G*). In addition to this apparent independence of the drug to varying *Pf*FKBP35 levels, and in agreement with earlier studies (*Bell et al., 1994*; *Harikishore et al., 2013b*; *Paul et al., 2015*), FK506 unfolds its antimalarial activity within the same cycle after treatment. This immediate activity of the drug stands in stark contrast to the delayed death-like phenotype of FKBP35[KO] parasites (compare *Figure 1E–G*), substantiating the notion that FK506 must target essential parasite factors other than *Pf*FKBP35.

To rule out that the timing of drug exposure is masking expected links between FK506 and *Pf*FKBP35, we tested whether administering FK506 at 30–36 hpi and 36–42 hpi (instead of 4–10 hpi) affects parasite survival in a *Pf*FKBP35-dependent manner (*Figure 4F*, *Figure 4—figure supplement 4*). However, we did not observe a change in FK506 susceptibility compared to FKBP35[WT] parasites, ruling out the possibility that FKBP35[KO] parasites were killed by FK506 before *Pf*FKBP35 levels were different in the two populations. Further, to account for a scenario in which the activity of FK506 on *Pf*FKBP35 is masked by a temporal delay between drug exposure and parasite killing, we induced the knock-out of *Pf*FKBP35 at 34–40 hpi (instead of 0–6 hpi) and allowed FKBP35[KO] parasites to proceed to G2, before exposing them to different FK506 concentrations. Despite these efforts, FKBP35[WT] and FKBP35[KO] parasites were equally sensitive to FK506 (*Figure 4—figure supplement 4*) and the calcineurin inhibitor cyclosporin A (*Figure 4—figure supplement 4*).

In addition to FK506, we evaluated the *Pf*FKBP35-specific inhibitor D44 (N-(2-ethylphenyl)–2-(3h-imidazo[4,5-B]pyridin-2-ylsulfanyl)acetamide) but, surprisingly, we failed to reproduce the previously reported dose–response effect (*Harikishore et al., 2013b*). We tested D44 from two suppliers using NF54/iKO-FKBP35, NF54 wild-type and K1 parasite strains using different assay setups, including the gold standard method that is based on the incorporation of tritium-labeled hypoxanthine into DNA of replicating cells (*Desjardins et al., 1979*). Still, we could not detect any activity of D44 on asexual replication of blood stage parasites (*Figure 4—figure supplement 4*).

Taken together, these results are in clear conflict with the perception that FK506 and D44 kill parasites by targeting *Pf*FKBP35 exclusively.

## Discussion

Members of the FKBP family are involved in regulating diverse cellular processes in eukaryotic as well as in prokaryotic cells (*Kang et al., 2008*; *Ünal and Steinert, 2015*). In contrast to most other eukaryotes, the human malaria parasite *P. falciparum* encodes a single FKBP only (*Kumar et al., 2005*). The enzymatic activity of *Pf*FKBP35, as well as the interactions of the protein with FKBP-targeting drugs, was subject to considerable in vitro drug-focused research. Besides demonstrating the drug's affinity to *Pf*FKBP35 by co-crystallization, these efforts uncovered that FK506 inhibits enzymatic activity of the protein's PPIase domain (*Monaghan and Bell, 2005*). Together with the antimalarial activity of FK506, these data offered compelling evidence for an essential but yet uncharacterized role of *Pf*FKBP35 during blood stage development of *P. falciparum*.

Here, we set out to investigate the role of *Pf*FKBP35 in parasite biology. Using an inducible knock-out approach, we show that *Pf*FKBP35 is indeed essential for intra-erythrocytic development of asexually replicating *P. falciparum* parasites and CETSA experiments confirmed high-affinity binding of FK506 to its *Pf*FKBP35 target. While the knock-out of *Pf*FKBP35 does not markedly affect transcription, FKBP35[KO] cells show reduced levels of ribosomal proteins and impaired protein synthesis on the global scale. Surprisingly however, these effects, as well as the consequent death during the late trophozoite/early schizont stage, only manifest in the cycle following knock-out induction without affecting cell cycle progression in the first generation (G1) or multiplication rates from G1 to G2. In contrast to the delayed death-like phenotype of FKBP35[KO] cells, FK506 unfolds its parasiticidal activity without a lag phase, that is, within the IDC of treatment (*Harikishore et al., 2013b*), and the dose–response relationship between FK506 and parasite death is irresponsive to altered *Pf*FKBP35 levels. Taken together, our data strongly indicate that the parasite-killing activity of FK506 is *Pf*FKBP35-independent.

Knocking out *Pf*FKBP35 early during intra-erythrocytic development has only minor effects on the parasite proteome in G1. Being relatively subtle, these changes may still point toward an important function of *Pf*FKBP35: factors associated with the nucleolus – the site of ribosome biogenesis – are

found at higher abundance in knock-out cells. In the next cycle, FKBP35[KO] parasites show a dramatic decrease of factors associated with ribosome- and translation-related processes and, likely a consequence thereof, globally reduced protein synthesis during trophozoite development (G2, 24–30 hpi). Importantly, at this point in time, FKBP35[KO] cells still show ordinary morphology and their transcriptional activity remains largely unaffected. The observed changes in the abundance of ribosomal factors must therefore result from transcription-independent activity of *Pf*FKBP35. While ribosomal components are present at higher levels in G1, they are drastically reduced in FKBP35[KO] cells in G2. These opposing effects likely result from *Pf*FKBP35 activity in a shared cellular process. Specifically, it is conceivable that low *Pf*FKBP35 levels inhibit ribosome maturation in G1, explaining the accumulation of (pre-) ribosomal factors. While the levels of functional ribosomes are sufficient to support continued translation in G1, ribosomes likely become limiting in the next generation (G2), which eventually results in stalled translation activity and concomitant parasite death at the onset of schizogony.

In contrast to the significant effects on the protein levels in G2, transcription of FKBP35[KO] cells remains largely unaffected by the loss of *Pf*FKBP35 and becomes apparent only during late trophozoite/early schizont development in G2 (30–36 hpi); that is, at a time during which cell cycle progression is slowed down in FKBP35[KO] parasites. In fact, at this stage the transcriptional signature of FKBP35[KO] parasites is dominated by the slowed progression through intra-erythrocytic development, rather than resulting from specific gene regulatory events. While this stalling is consistent with the observed effects on ribosome homeostasis and protein synthesis, it demonstrates that, in contrast to certain FKBP family members in other organisms (*Ghartey-Kwansah et al., 2018*), *Pf*FKBP35 does not have apparent roles in transcriptional control.

Regarding the well-characterized and fast-acting parasiticidal activity of FK506 on *P. falciparum*, it was somewhat surprising to observe a delayed death-like phenotype for *Pf*FKBP35 knock-out parasites (*Harikishore et al., 2013b*). We hence set out to investigate unifying explanations. The delayed death of FKBP35[KO] parasites is most easily explained by residual activity of *Pf*FKBP35 proteins synthesized prior to inducing the genomic knock-out of *fkbp35*. In such a scenario, low *Pf*FKBP35 in FKBP35[KO] would be sufficient to allow completion of the IDC, before eventually dropping below a level that is required for supporting parasite survival in the next cycle. Despite intense efforts, we failed to confirm this hypothesis and our data strongly favors an alternative scenario in which FK506 kills parasites in a *Pf*FKBP35-idenpendent manner. Aside from the fact that FK506 exerts its parasiticidal activity within the same IDC, neither the conditional knock-out, knock-down, nor the overexpression of *Pf*FKBP35 altered the parasite's sensitivity to FK506; and this irrespective of the profoundly deregulated *Pf*FKBP35 levels in the respective mutant lines (*Figure 4F*). For instance, administering FK506 36 hr post induction of the knock-out in early ring stages (36–42 hpi) – that is, at a time point where FKBP35[KO] cells are only left with about 5% of the regular *Pf*FKBP35 levels (19-fold reduction compared to FKBP35[WT] parasites, *Figure 2A*) – does not affect the dose–response of parasites to the drug. Further, our attempts to demonstrate that the timing of drug exposure is masking the expected links between FK506 and its *Pf*FKBP35 drug target failed (compare *Figure 4A* and *Figure 4—figure supplement 4*). Our data is thus incompatible with the notion that the parasite-killing activity of FK506 results from inhibiting the essential function of *Pf*FKBP35.

Since binding of FK506 to *Pf*FKBP35 alone cannot explain the drug's antimalarial activity, we aimed to identify additional FK506 targets. CETSA performed on protein lysates identified a number of parasite factors binding to FK506. Compared to the *Pf*FKBP35/FK506 pair, however, these interactions are of considerably lower affinity. The adenosine-methyltransferase MT-A70 (PF3D7_0729500) and the pre-mRNA-splicing factor CEF1 (PF3D7_1033600), for instance, were stabilized by FK506 with EC50 concentrations of 2.0 and 3.1 μM, respectively. While this affinity is relatively poor compared to the strong interaction formed between *Pf*FKBP35 and FK506, one should be cautious about designating the binding of FK506 to these proteins as irrelevant '*off-target*' effects. In fact, their inhibition may hypothetically mask the effects of FK506 on *Pf*FKBP35 – a possibility that shall briefly be discussed in the following section.

Assuming that FK506 is inhibiting the essential function of *Pf*FKBP35, one would expect parasites to show a delayed death-like phenotype similar to that observed in FKBP35[KO] parasites. The presence and inhibition of other essential targets may, however, mask such an effect. Consistent with this possibility, FK506 kills *P. falciparum* parasites at 1.9 μM (± 0.2 μM s.d.), that is, at concentrations similar to those required for stabilizing MT-A70 and CEF1 under thermal challenge in CETSA.

However, numerous attempts to prove that FK506 indeed inhibits essential *Pf*FKBP35 functions failed: neither knocking out *Pf*FKBP35 at different time points nor adding FK506 to different stages of FKBP35**KO** parasites could provide evidence for a *Pf*FKBP35-dependent effect. This indicates that either the binding of FK506 does not interfere with the essential role of *Pf*FKBP35, or that *Pf*FKBP35 is inhibited only at high FK506 concentrations that also inhibit other essential factors. In this context, it is worth mentioning that parasites remain unaffected by a highly efficient knock-down of endogenous *Pf*FKBP35 (observed with NF54/iKD-FKBP35; compare *Figure 1—figure supplement 1*), which demonstrates that very low protein levels are sufficient to maintain the essential function of *Pf*FKBP35. It is thus conceivable that high FK506 concentrations are required to fully inhibit *Pf*FKBP35 – a view that is further supported by data obtained with the live cell CETSA approach discussed hereafter.

Besides identifying direct binding of drugs to their target, live cell CETSA allows capturing indirect effects on factors that act in the same pathway or in the same protein complex as the drug-bound target (*Dziekan et al., 2020*). While this approach confirmed the high-affinity binding of FK506 to *Pf*FKBP35, it also revealed that more than 50 ribosomal proteins share a very prominent CETSA signature: first, they are thermally stabilized by FK506 within a narrow concentration window of the drug. Second, higher levels of FK506 increase the solubility of these ribosomal components also at 37°C. These results indicate that the FK506/*Pf*FKBP35 pair interacts with ribosomal complexes and that FK506 is able to induce the dissociation of these ribosomal structures under physiological conditions. Importantly, and in contrast to the binding of FK506 to *Pf*FKBP35, this effect on ribosomes is only observed at high concentrations of the drug. Together, these data likely explain the discrepancy observed between the high-binding affinity of FK506 for *Pf*FKBP35 and the low or even absent effect of the drug on *Pf*FKBP35 function in blood stage parasites.

In line with our data, previous biochemical approaches revealed potential links between *Pf*FKBP35 and ribosomes. Using co-immunoprecipitation, Leneghan and colleagues revealed that *Pf*FKBP35 is interacting with 480 proteins, of which 31 are ribosomal factors (*Leneghan and Bell, 2015*). Indeed, FKBP family members are essentially involved in controlling ribosomes in other systems. The human FKBP variant *Hs*FKBP25, for instance, is recruited to the pre-ribosome where its chaperone activity appears to be required for the assembly of the ribosomal large subunit (*Gudavicius et al., 2014*). While the FKBP variant of yeast, Fpr4, interacts with the ribosome biogenesis factor Nop53 (*Sydorskyy et al., 2005*), the *Escherichia coli* trigger factor (TF) is part of the ribosomal complex and crucial for the folding of nascent proteins (*Ferbitz et al., 2004*; *Callebaut and Mornon, 1995*). Considering the marked reduction of ribosomal proteins in FKBP35**KO** parasites, it is tempting to speculate that *Pf*FKBP35 is chaperoning ribosome biogenesis similar to Fpr4 and FKBP25 in *Saccharomyces cerevisiae* and *human* cells, respectively.

The fact that *Pf*FKBP35 localizes to distinct nuclear foci and its depletion causes dysregulation of ribosomes points toward a role within the nucleolus – the nuclear compartment driving ribosome biogenesis (*Boisvert et al., 2007*). Using IFAs, we found that *Pf*FKBP35 does not co-localize with the nucleolar marker fibrillarin (*Rodriguez-Corona et al., 2015*; *Figure 4—figure supplement 5*). However, its localization appears to be intertwined with that of this rRNA methyltransferase. Specifically, *Pf*FKBP35 is mostly found in close vicinity to fibrillarin-rich regions. Nucleoli in eukaryotic cells generally consist of two or, in higher eukaryotes, three sub-compartments. Of those, the granular component (GC) represents the site of pre-ribosome assembly and encompasses fibrillarin-enriched central regions of nucleoli (*Thiry and Lafontaine, 2005*). The spatial correlation observed between *Pf*FKBP35 and fibrillarin may thus indicate localization to different sub-nucleolar compartments, with *Pf*FKBP35 occupying the GC area of the nucleolus.

In summary, we demonstrate that limiting *Pf*FKBP35 levels are lethal to *P. falciparum* and result in a delayed death-like phenotype that is characterized by perturbed ribosome homeostasis and defective protein synthesis, which is likely linked to the reduced biogenesis of functional ribosomes in the nucleolus. We further found no evidence that FKBP-binding drugs, including FK506, exert their parasiticidal activity in a *Pf*FKBP35-dependent manner, urging strong caution for the future development of FKBP-targeting antimalarials, especially when based on FK506 and structural derivatives thereof. In addition to revealing first insights into the essential function of *Pf*FKBP35 in ribosome homeostasis of *P. falciparum*, we are convinced that the presented data offer valuable information for target-based efforts in malaria drug discovery.

## Methods

### Parasite culture

*P. falciparum* culture and synchronization were performed as described (*Lambros and Vanderberg, 1979*; *Trager and Jenson, 1978*). Parasites were cultured in AB+ or B+ human red blood cells (Blood Donation Center, Zürich, Switzerland) at a hematocrit of 5% in culture medium containing 10.44 g/L RPMI-1640, 25 mM HEPES, 100 µM hypoxanthine, 24 mM sodium bicarbonate, 0.5% AlbuMAX II (Gibco #11021-037), 0.1 g/L neomycin, and 2 mM choline chloride (Sigma #C7527). To achieve stabilization of DD-tagged proteins in the cell lines NF54/iKD-FKBP35 and NF54/iOE-FKBP35, parasites were cultured in the presence of 625 nM Shield-1 (*Armstrong and Goldberg, 2007*; *Banaszynski et al., 2006*). To induce the DiCre-mediated recombination of *loxP* sites in NF54/iKO-FKBP35 parasites, cultures were split and treated for 4 hr with 100 nM rapamycin or the corresponding volume DMSO, which served as vehicle control, giving rise to FKBP35**KO** and FKBP35**WT** populations, respectively (*Collins et al., 2013*). Cultures were gassed with 3% $O_2$, 4% $CO_2$, and 93% $N_2$ and incubated in an airtight incubation chamber at 37°C.

### Cloning of transfection constructs

CRISPR/Cas9-based gene editing of the NF54 parasites was performed using a two-plasmid approach as previously described (*Brancucci et al., 2017*; *Filarsky et al., 2018*). This system is based on co-transfection of a plasmid that contains the expression cassettes for the Cas9 enzyme, the single-guide RNA (sgRNA) and the blasticidin deaminase (BSD) resistance marker (pBF-gC), and a pD-derived donor plasmid that contains the template for the homology-directed repair of the Cas9-induced DNA double-strand break (*Figure 1—figure supplement 2*; *Filarsky et al., 2018*).

The plasmids pBF-gC_FKBP-3′, pBF-gC_FKBP-5′, and pBF-gC_P230p targeting the 3′ or 5′ end of *fkbp35* (PF3D7_1247400) or the *p230p* (PF3D7_0208900) locus (*Knuepfer et al., 2017*), respectively, were generated by ligation of two annealed oligonucleotides (gRNA_top and gRNA_bottom) into the BsaI-digested pBF-gC backbone (*Filarsky et al., 2018*) using T4 DNA ligase (New England Biolabs).

The donor plasmid pD_FKBP35-iKO was generated by assembling six DNA fragments in a Gibson reaction (*Gibson et al., 2009*) using (i) the plasmid backbone amplified by polymerase chain reaction (PCR) from pUC19 (primers PCRa_F/PCRa_R) (*Filarsky et al., 2018*), (ii/iii) the 5′ and 3′ homology regions amplified from genomic DNA (primers PCRb_F/PCRb_R and PCRc_F/PCRc_R, respectively), (iv) the *gfp* sequence amplified from pHcamGDV1-GFP-DD (primers PCRd_F/PCRd_R) (*Filarsky et al., 2018*), (v) the HRPII terminator amplified from genomic DNA (primers PCRe_F/PCRe_R), and (vi) the *fkbp35* sequence containing an artificial *loxPint* (*Jones et al., 2016*) amplified from a *P. falciparum* codon-optimized synthetic sequence (primers PCRf_F/PCRf_R) ordered from GenScript.

The donor plasmid pD_FKBP35-iKD was generated by assembling five DNA fragments in a Gibson reaction (*Gibson et al., 2009*) using (i) the plasmid backbone amplified from pUC19 (primers PCRa_F/PCRa_R) (*Filarsky et al., 2018*), (ii/iii) the 5′ and 3′ homology regions amplified from genomic DNA (primers PCRg_F/PCRg_R and PCRh_F/PCRh_R, respectively), (iv) the *gfp* sequence amplified from pHcamGDV1-GFP-DD (primers PCRi_F/PCRi_R) (*Filarsky et al., 2018*), and (v) the destabilization domain (*dd*) sequence amplified from pHcamGDV1-GFP-DD (primers PCRj_F/PCRj_R) (*Filarsky et al., 2018*).

The donor plasmid pD_FKBP35-iOE was generated by assembling three DNA fragments in a Gibson reaction (*Gibson et al., 2009*) using (i) the NheI/PstI-digested plasmid pkiwi003 (*Ashdown et al., 2020*), (ii) the *fkbp35* coding sequence amplified from genomic DNA (primers PCRk_F/PCRk_R), and (iii) the *gfp-dd* sequence amplified from pHcamGDV1-GFP-DD (primers PCRl_F/PCRl_R) (*Filarsky et al., 2018*).

Oligonucleotides are listed in *Supplementary file 4*.

### Transfection and selection of gene-edited parasites

*P. falciparum* transfection using the CRISPR/Cas9 two-plasmid approach was performed as described previously (*Filarsky et al., 2018*). Briefly, 50 µg of each plasmid (pBF-gC_FKBP-3′ and pD_FKBP35-iKO, pBF-gC_FKBP-5′ and pD_FKBP35-iKD, pBF-gC_P230p, and pD_FKBP35-iOE) were co-transfected into an NF54::DiCre line (*Tibúrcio et al., 2019*), in which *ap2-g* was tagged with the fluorophore mScarlet as described previously (*Thommen et al., 2022*). Transgenic parasites were selected with 2.5 µg/mL blasticidin-S-hydrochloride, which was added 24 hr after transfection for 10 d. Transgenic

populations were usually obtained 2–3 wk after transfection and correct editing of the modified loci was confirmed by PCR on gDNA (*Figure 1—figure supplement 1*, *Figure 1—figure supplement 2*, *Figure 4—figure supplement 3*). Primer sequences used for these PCRs are listed in *Supplementary file 4*. Clonal parasite lines were obtained by limiting dilution cloning (*Thomas et al., 2016*).

## DNA content analysis

NF54/iKO-FKBP35 parasites were fixed in 4% formaldehyde/0.0075% glutaraldehyde for 30 min at room temperature (RT) and washed three times in PBS and stored at 4°C. Samples were collected from three independent biological replicates. RNA was digested by incubation with 0.1 mg/mL RNAse A in PBS containing 0.1% Triton-X100 for 15 min at RT. Nuclei were stained using 1X SYBR Green 1 DNA stain (Invitrogen S7563) for 20 min and washed three times in PBS. The fixation step quenched the GFP signal detected by the flow cytometer (*Figure 1—figure supplement 3*). SYBR Green intensity was measured using a MACS Quant Analyzer 10 and analyzed using the FlowJo_v10.6.1 software.

## Western blotting

Saponin lysis of 500 µL infected red blood cells at 3–5% parasitemia was performed by incubation for 10 min on ice in 3 mL ice-cold 0.15% saponin in PBS. The parasite pellets were washed two times in ice-cold PBS and resuspended in Laemmli buffer (62.5 mM Tris base, 2% SDS, 10% glycerol, and 5% 2-mercaptoethanol). Proteins were separated on 4–12% Bis-Tris gels (Novex, QIAGEN) using MOPS running buffer (Novex, QIAGEN). Proteins were then transferred to a nitrocellulose membrane (GE Healthcare #106000169), which was blocked with 5% milk powder in PBS/0.1% Tween (PBS-T) for 1 hr. The membrane was probed using the primary antibodies mAb mouse α-GFP (1:1000, Roche Diagnostics #11814460001), mAb mouse α-*Pf*GAPDH (1:20,000) (*Daubenberger et al., 2003*), or mAb mouse α-puromycin (1:5000, Sigma MABE343) diluted in blocking solution. After 2 hr incubation, the membrane was washed five times in PBS-T before it was incubated with the secondary antibody goat α-mouse IgG (H&L)-HRP (1:10,000, GE Healthcare #NXA931). After 1 hr incubation, the membrane was washed five times and the signal was detected using the chemiluminescent substrate SuperSignal West Pico Plus (Thermo Scientific, REF 34580) and the imaging system Vilber Fusion FX7 Edge 17.10 SN. Nitrocellulose membranes were stripped by incubating for 10 min in a stripping buffer (1 g/L SDS, 15 g/L glycine, 1% Tween-20, pH 2.2) and washed three times in PBS-T before blocking and re-probing.

## Drug dose–response experiments

Dose–response relationships were assessed by exposing synchronous ring stage parasites at 0.5% parasitemia and 1.25% hematocrit to 12 drug concentrations using a two-step serial dilution in 96-well plates (Corning, 96-well cell culture plate, flat bottom, REF 3596) in three independent biological replicates. The plates were gassed and incubated for 48 hr at 37°C. FK506 (MedChemExpress HY-13756), D44 (ChemBridge 7934155 and ChemDiv 7286-2836) and cyclosporin A (Sigma 30024) stocks of 10 mM were prepared in DMSO and working solutions were prepared immediately prior to starting the experiment. To measure parasite survival using flow cytometry, cultures were stained in 1X SYBR Green 1 DNA stain (Invitrogen S7563) in a new 96-well plate (Corning Incorporated, 96-well cell culture plate, round bottom, REF 3788) and incubated in the dark for 20 min. The samples were washed once in PBS. Using a MACS Quant Analyzer 10, 50,000 events per condition were measured and analyzed using the FlowJo_v10.6.1 software. Infected RBCs were identified based on SYBR Green 1 DNA stain intensity. The gating strategy is shown in *Figure 1—figure supplement 3*. To quantify parasite survival based on hypoxanthine incorporation into DNA, compounds were dissolved in DMSO (10 mM), diluted in hypoxanthine-free culture medium and titrated in duplicates over a 64-fold range (six-step twofold dilutions) in 96-well plates. 100 µL asexual parasite culture (prepared in hypoxanthine-free medium) were added to each well and mixed with the compound to obtain a final hematocrit of 1.25% and a final parasitemia of 0.3%. After incubation for 48 hr, 0.25 µCi of [3H]-hypoxanthine was added per well and plates were incubated for an additional 24 hr. Parasites were then harvested onto glass-fiber filters using a Microbeta FilterMate cell harvester (PerkinElmer, Waltham, USA) and radioactivity was quantified using a MicroBeta2 liquid scintillation counter (PerkinElmer). The results were recorded and expressed as a percentage of the untreated controls. Curve fitting and

IC50 calculations were performed using a nonlinear, four-parameter regression model with variable slope (Graph Pad Prism, version 8.2.1).

## Fluorescence microscopy

To localize *Pf*FKBP35, NF54/iKD-FKBP35 parasites were fixed in 4% formaldehyde/0.0075% glutaraldehyde for 30 min at RT and washed three times in PBS. The fixed cells were permeabilized by incubation with 0.1% Triton-X100-containing PBS for 15 min. After two washing steps in PBS, the cells were incubated in a blocking/aldehyde quenching solution (3% BSA in PBS complemented with 50 mM ammonium chloride). The samples were incubated with the primary antibodies rabbit α-GFP (1:400, Abcam ab6556) and mAb mouse α-fibrillarin (1:100, Santa Cruz Biotechnology sc-166021) for 1 hr in 3% BSA in PBS and washed three times in PBS. Subsequently, the samples were incubated with the secondary antibodies goat α-rabbit IgG (H&L) Alexa Fluor 488 (1:250, Invitrogen A11008) and goat α-mouse IgG (H&L) Alexa Fluor 594 (1:250, Invitrogen A11032) for 1 hr and washed three times, before they were mixed with Vectashield containing DAPI (Vector laboratories, H-1200), and mounted on a microscopy slide. Slides were imaged using a Leica THUNDER 3D Assay imaging system.

## Proteomics

Parasite cultures were split at 0–6 hpi in G1 and treated for 4 hr with 100 nM rapamycin. Saponin lysis of paired FKBP35$^{WT}$ and FKBP35$^{KO}$ populations at 36–42 hpi in G1 and 24–30 hpi in G2 was performed as described above. Samples were collected in three independent biological replicates. The washed parasite pellet was snap frozen in liquid nitrogen. After thawing, parasites were resuspended in 25 µL of lysis buffer (5% SDS, 100 mM tetraethylammonium bromide [TEAB], 10 mM tris(2-carboxyethyl) phosphin [TCEP], pH 8.5) and sonicated (10 cycles, 30 s on/off at 4°C, Bioruptor, Diagnode). Lysates were subsequently reduced for 10 min at 95°C. Samples were then cooled down to RT and 0.5 µL of 1 M iodoacetamide was added to the samples. Cysteine residues were alkylated for 30 min at 25°C in the dark. Digestion and peptide purification was performed using S-trap technology (Protifi) according to the manufacturer's instructions. In brief, samples were acidified by addition of 2.5 µL of 12% phosphoric acid (1:10) and then 165 µL of S-trap buffer (90% methanol, 100 mM TEAB, pH 7.1) was added to the samples (6:1). Samples were briefly vortexed and loaded onto S-trap micro spin-columns (Protifi) and centrifuged for 1 min at 4000 × *g*. Flow-through was discarded and spin-columns were then washed three times with 150 µL of S-trap buffer (each time samples were centrifuged for 1 min at 4000 × *g* and flow-through was removed). S-trap columns were then moved to the clean tubes and 20 µL of digestion buffer (50 mM TEAB pH 8.0) and trypsin (at 1:25 enzyme to protein ratio) were added to the samples. Digestion was allowed to proceed for 1 hr at 47°C. Then, 40 µL of digestion buffer were added to the samples and the peptides were collected by centrifugation at 4000 × *g* for 1 min. To increase the recovery, S-trap columns were washed with 40 µL of 0.2% formic acid in water (400 × *g*, 1 min) and 35 µL of 0.2% formic acid in 50% acetonitrile. Eluted peptides were dried under vacuum and stored at –20°C until further analysis.

Peptides were resuspended in 0.1% aqueous formic acid and peptide concentration was adjusted to 0.25 µg/µL. Then, 1 µL of each sample was subjected to LC-MS/MS analysis using an Orbitrap Elicpse Tribrid Mass Spectrometer fitted with an Ultimate 3000 nanosystem (both from Thermo Fisher Scientific) and a custom-made column heater set to 60°C. Peptides were resolved using a RP-HPLC column (75 µm × 30 cm) packed in-house with C18 resin (ReproSil-Pur C18-AQ, 1.9 µm resin; Dr. Maisch GmbH) at a flow rate of 0.3 µL/min. The following gradient was used for peptide separation: from 2% buffer B to 12% B over 5 min, to 30% B over 40 min, to 50% B over 15 min, to 95% B over 2 min followed by 11 min at 95% B then back to 2% B. Buffer A was 0.1% formic acid in water and buffer B was 80% acetonitrile, 0.1% formic acid in water.

The mass spectrometer was operated in DDA mode with a cycle time of 3 s between master scans. Each master scan was acquired in the Orbitrap at a resolution of 240,000 FWHM (at 200 m/z) and a scan range from 375 to 1600 m/z followed by MS2 scans of the most intense precursors in the linear ion trap at 'Rapid' scan rate with isolation width of the quadrupole set to 1.4 m/z. Maximum ion injection time was set to 50 ms (MS1) and 35 ms (MS2) with an AGC target set to 1e6 and 1e4, respectively. Only peptides with charge state 2–5 were included in the analysis. Monoisotopic precursor selection (MIPS) was set to Peptide, and the Intensity Threshold was set to 5e3. Peptides were fragmented by

higher-energy collisional dissociation (HCD) with collision energy set to 35%, and one microscan was acquired for each spectrum. The dynamic exclusion duration was set to 30 s.

The acquired raw files were imported into the Progenesis QI software (v2.0, Nonlinear Dynamics Limited), which was used to extract peptide precursor ion intensities across all samples applying the default parameters. The generated mgf-file was searched using MASCOT against a *Plasmodium falciparum* (isolate 3D7) database (UniProt, 11.2019) and 392 commonly observed contaminants using the following search criteria: full tryptic specificity was required (cleavage after lysine or arginine residues, unless followed by proline); three missed cleavages were allowed; carbamidomethylation (C) was set as fixed modification; oxidation (M) and acetyl (Protein N-term) were applied as variable modifications; mass tolerance of 10 ppm (precursor) and 0.6 Da (fragments). The database search results were filtered using the ion score to set the false discovery rate (FDR) to 1% on the peptide and protein level, respectively, based on the number of reverse protein sequence hits in the dataset. Quantitative analysis results from label-free quantification were processed using the SafeQuant R package v.2.3.2 (*Ahrné et al., 2016*) to obtain peptide relative abundances. This analysis included global data normalization by equalizing the total peak/reporter areas across all LC-MS runs, data imputation using the knn algorithm, summation of peak areas per protein and LC-MS/MS run, followed by calculation of peptide abundance ratios. Only isoform-specific peptide ion signals were considered for quantification. To meet additional assumptions (normality and homoscedasticity) underlying the use of linear regression models and *t*-tests, MS-intensity signals were transformed from the linear to the log-scale. The summarized peptide expression values were used for statistical testing of between condition differentially abundant peptides. Here, empirical Bayes moderated *t*-tests were applied, as implemented in the R/Bioconductor limma package (*Ritchie et al., 2015*). The resulting per protein and condition comparison p-values were adjusted for multiple testing using the Benjamini–Hochberg method.

Gene ontology enrichment analysis was performed using the PANTHER Overrepresentation Test with the correction method 'false discovery rate,' the test type 'Fisher's exact' and the annotation dataset 'PANTHER GO-Slim' (*Mi et al., 2021*).

## IPP complementation

Parasite cultures were split at 0–6 hpi in G1 and treated for 4 hr with 100 nM rapamycin. FKBP35**KO** and FKBP35**WT** populations were split again and supplemented either with 200 μM isopentenyl pyrophosphate (IPP) (Sigma I0503) or the corresponding volume $H_2O$. The respective culture medium was replaced daily. Parasites treated with 156 ng/mL doxycycline (Sigma D9891) were used to confirm that IPP complementation is able to rescue apicoplast-deficient parasites. DNA content analysis was performed as described above.

## Heat shock susceptibility testing

Parasites cultures were split at 0–6 hpi in G1 and treated for 4 hr with 100 nM rapamycin. While control populations were always incubated at 37°C, test populations were incubated at 40°C for 6 hr starting at 24–30 hpi. Parasitemia was measured in G2 using flow cytometry as described above.

## Surface sensing of translation (SUnSET)

Parasite cultures were split at 0–6 hpi in G1 and treated for 4 hr with 100 nM rapamycin. At 24–30 hpi in G2, the cultures were incubated with 1 μg/mL puromycin (Sigma P8833) for 1 hr at 37°C (*Schmidt et al., 2009*; *McLean and Jacobs-Lorena, 2017*). Saponin lysis and western blotting were performed as described above.

## Cellular thermal shift assay (CETSA)

The mass spectrometry-based isothermal dose–response cellular thermal shift assay (ITDR-MS-CETSA) was performed according to the protocol developed by Dziekan and colleagues (*Dziekan et al., 2020*). Briefly, live parasites ($10^7$ MACS-purified trophozoites per condition) were exposed to an FK506 (MedChemExpress HY-13756) concentration gradient (100 μM to 1.5 nM) and a DMSO control for 1 hr. Subsequently, the parasites were washed in PBS and exposed to the denaturing temperatures 50°C, 55°C, and 60°C, as well as to a non-denaturing temperature (37°C) for 3 min, before they were cooled down to 4°C for 3 min. Afterward, the cells were lysed by resuspending in a lysis buffer (50 mM

HEPES pH 7.5, 5 mM β-glycerophosphate, 0.1 mM $Na_3VO_4$, 10 mM $MgCl_2$, 2 mM TCEP, and EDTA-free protease inhibitor cocktail [Sigma]), three freeze/thaw cycles, and mechanical shearing using a syringe with a 31-gauge needle. Subsequently, insoluble proteins were pelleted by centrifugation (20 min at 4°C and 20,000 × $g$) and the soluble fraction was collected and subjected to protein digestion, reduction, alkylation, and TMT10 labeling followed by LC/MS analysis as described (*Dziekan et al., 2020*).

To prepare the protein lysate, saponin-treated parasites were lysed in the aforementioned buffer by three freeze/thaw cycles and mechanical shearing. After centrifugation, soluble proteins were exposed to an FK506 concentration gradient (200 µM to 3 nM) and a DMSO control for 1 min before the thermal challenge was performed as described above. Afterward, the soluble protein fraction was isolated and prepared for by LC/MS analysis as described (*Dziekan et al., 2020*).

Data analysis was performed using the Proteome Discoverer 2.1 software (Thermo Fisher Scientific) and the R package 'mineCETSA' (v 1.1.1). Only proteins identified by at least three peptide spectrum matches (PSMs) were included in the analysis (*Dziekan et al., 2020*).

## Transcriptomics

### Sample preparation

The RNA of 500 µL iRBCs at 3–5% parasitemia was isolated using 3 mL TRIzol reagent (Invitrogen) followed by phenol-chloroform phase separation. The aqueous phase was purified using the Direct-zol RNA MicroPrep kit (Zymo). Samples were collected from three biological replicates. Stranded RNA sequencing libraries were prepared using the Illumina TruSeq stranded mRNA library preparation kit (REF 20020594) according to the manufacturer's protocol. The library was sequenced in 100 bp paired-end reads on an Illumina NextSeq 2000 sequencer using the Illumina NextSeq 2000 P3 reagents (REF 20040560), resulting in 5 million reads on average per sample.

### RNA-seq data analysis

Fastqc was run on all raw fastq files. Then, each file was aligned with hisat2 (version 2.0.5) (*Kim et al., 2019*) against the 3D7 reference genome (PlasmoDB v58) (*Aurrecoechea et al., 2009*) in which the GFP and the recoded FBPK35 sequences were added at the end of chromosome 12. Read counts for each gene and for each sample were then counted using featureCount (gff file from PlasmoDB v58 modified to include the added GFP and FBPK35). TPM (transcripts per kilobase million) were then calculated from the raw count matrix.

Principal component analysis (PCA) was performed using the R (version 4.2.1) function prcomp using log-transformed TPMs. PCA revealed that the rapamycin-treated replicate B of TP 3 ('3B.R') was a clear outlier and is different from any other replicate at any time point (*Figure 3—figure supplement 1*). For this reason, it was excluded from further analysis. The PCA was done again without it and the other samples show the same pattern as the initial PCA (*Figure 3—figure supplement 1*).

DESeq2 (version 1.36) (*Love et al., 2014*) was used for normalization and differential expression analysis. Paired differential expression tests were done for each time point using DESeq2 LRT (likelihood ratio test) using the full model ~ Treatment + Replicate (Treatment is DMSO or Rapamycin and Replicate is the clone, added as the data is paired) and the reduced model ~Replicate. From this paired test and the average log fold-change calculated from the TPMs, normalized volcano plots were made using the R package EnhancedVolcano (version 1.14) (*Blighe et al., 2022*).

### Cell cycle analysis

Spearman correlation between normalized scaled read counts for each sample for all genes (as one vector) and each time point of the reference microarray time course published by *Bozdech et al., 2003* (3D7 smoothed, retrieved from PlasmoDB) were tested. Spearman's coefficients rho for each time point of this RNA-seq time course and each time point of the reference dataset were plotted using ggcorrplot (version 0.1.3; *Kassambara, 2022*). For each time point of this RNA-seq dataset, the rho coefficients were then plotted against the time points of the reference dataset and the FKBP35[WT] and FKBP35[KO] data were fitted separately to the linearized sinusoidal $A * sin\left(\frac{2\pi}{53} time\right) + B * cos\left(\frac{2\pi}{53} time\right) + C$ using R lm function. The time point matching best to the FKBP35[WT] and FKBP35[KO] samples for each time point of the RNA-seq time course was taken as the

maximum of this fitted function. Both fitted functions were F-tested (using R function var.test using both fitted model as applied to the FKBP35$^{KO}$ data) to test whether the FKBP35$^{WT}$ and FKBP35$^{KO}$ data correlated with the reference dataset (*Bozdech et al., 2003*) in a significantly different way, which would indicate a cell cycle shift if statistically significant. Only the last time point showed such a significant difference.

## Acknowledgements

We thank Vera Mitesser and Ron Dzikowski for guidance on the SUnSET experiments and David Fidock for providing the parasite line Dd2-B2 used for the attempted resistance selection. We are grateful to the Polyomics facility at the University of Glasgow for processing the samples for RNA-seq. This work was supported by the Swiss National Science Foundation (grant number 310030_200683), the Wolfson Merit Royal Society Award, the Wellcome Trust Investigator Award 110166 and Wellcome Trust Center Award 104111, and the Singapore Ministry of Education (grant number MOE-T2EP30120-0015).

## Additional information

### Funding

| Funder | Grant reference number | Author |
| --- | --- | --- |
| Schweizerischer Nationalfonds zur Förderung der Wissenschaftlichen Forschung | 310030_200683 | Nicolas MB Brancucci |
| Wellcome Trust | 110166 | Matthias Marti |
| Wellcome Trust | 104111 | Matthias Marti |
| Ministry of Education - Singapore | T2EP30120-0015 | Zbynek Bozdech |

The funders had no role in study design, data collection and interpretation, or the decision to submit the work for publication. For the purpose of Open Access, the authors have applied a CC BY public copyright license to any Author Accepted Manuscript version arising from this submission.

### Author contributions

Basil T Thommen, Data curation, Formal analysis, Investigation, Visualization, Methodology, Writing – original draft, Writing – review and editing; Jerzy M Dziekan, Data curation, Formal analysis, Supervision, Investigation, Methodology; Fiona Achcar, Data curation, Formal analysis, Visualization; Seth Tjia, Armin Passecker, Christin Gumpp, Investigation, Methodology; Katarzyna Buczak, Data curation, Formal analysis, Methodology; Alexander Schmidt, Supervision, Methodology; Matthias Rottmann, Resources, Supervision; Christof Grüring, Supervision, Writing – review and editing; Matthias Marti, Resources, Supervision, Funding acquisition, Writing – review and editing; Zbynek Bozdech, Resources, Supervision, Funding acquisition, Methodology, Writing – review and editing; Nicolas MB Brancucci, Conceptualization, Resources, Data curation, Formal analysis, Supervision, Funding acquisition, Investigation, Visualization, Methodology, Project administration, Writing – review and editing

### Author ORCIDs

Basil T Thommen https://orcid.org/0000-0001-7655-2440
Fiona Achcar http://orcid.org/0000-0001-8792-7615
Alexander Schmidt http://orcid.org/0000-0002-3149-2381
Matthias Rottmann https://orcid.org/0000-0002-2207-7899
Zbynek Bozdech http://orcid.org/0000-0002-9830-8446
Nicolas MB Brancucci https://orcid.org/0000-0003-0655-3266

Joint Public Review: https://doi.org/10.7554/eLife.86975.4.sa1

Author Response https://doi.org/10.7554/eLife.86975.4.sa2

## Additional files

### Supplementary files
• Supplementary file 1. Mass spectrometry data comparing the proteomes of FKBP35[KO] and FKBP35[WT] parasites.

• Supplementary file 2. RNA-seq data comparing the transcriptomes of FKBP35[KO] and FKBP35[WT] parasites.

• Supplementary file 3. CETSA data showing the relative abundance of soluble proteins under non-denaturing and denaturing conditions in response to increasing FK506 concentrations.

• Supplementary file 4. Oligonucleotides used in this study.

• MDAR checklist

### Data availability
The mass spectrometry proteomics data have been deposited to the ProteomeXchange Consortium repository with the dataset identifier PXD039018. The RNA-sequencing data are available in the Sequence Read Archive (SRA) via the accession number PRJNA914079.

The following datasets were generated:

| Author(s) | Year | Dataset title | Dataset URL | Database and Identifier |
|---|---|---|---|---|
| Thommen BT, Dziekan JM, Achcar F, Tjia S, Passecker A, Buczak K, Gumpp C, Schmidt A, Rottmann M, Grüring C, Marti M, Bozdech Z, Brancucci NMB | 2022 | FKBP35 secures ribosome homeostasis in *Plasmodium falciparum* | https://www.ncbi.nlm.nih.gov/bioproject/PRJNA914079 | NCBI BioProject, PRJNA914079 |
| Thommen BT, Dziekan JM, Achcar F, Tjia S, Passecker A, Buczak K, Gumpp C, Schmidt A, Rottmann M, Grüring C, Marti M, Bozdech Z, Brancucci NMB | 2022 | FKBP35 secures ribosome homeostasis in *Plasmodium falciparum* | http://www.ebi.ac.uk/pride/archive/projects/PXD039018 | PRIDE, PXD039018 |

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
