## [Editor Report · eLife assessment]

FKBP35 is the only FK506-binding protein present in the malaria-causing parasite *Plasmodium falciparum*, and has been considered a promising drug target due to its high affinity to the macrolide compound FK506, an immunosuppressant with antiplasmodial activity. This study demonstrates the essentiality of FKBP35 in parasite growth, based on **compelling** genetic evidence. The data also suggest that FK506 may exert its antimalarial activity through FKBP35-independent mechanisms that have not yet been characterized. This **important** study will be of interest to scientists working on the parasite biology and antimalarial drug development.

---

## [Referee Report · Joint Public Review]

In this study, the authors investigate the biological function of the FK506-binding protein FKBP35 in the malaria-causing parasite *Plasmodium falciparum*. Like its homologs in other organisms, PfFKBP35 harbors peptidyl-prolyl isomerase and chaperoning activities, and has been considered a promising drug target due to its high affinity to the macrolide compound FK506. However, PfFKBP35 has not been validated as a drug target using reverse genetics, and the link between PfFKBP35-interacting drugs and their antimalarial activity remains elusive. The manuscript addresses the biological function of PfFKBP35 and the antimalarial activity of FK506.

The authors combine conditional genome editing, proteomics and transcriptomics analysis to investigate the effects of FKBP35 depletion in *P. falciparum*. The work is very well performed and clearly described. The data provide conclusive evidence that FKBP35 is essential for P. falciparum blood stage growth. Conditional knockout of PfFKBP35 leads to a delayed death-like phenotype, associated with defects in ribosome maturation as detected by quantitative proteomics and stalling of protein synthesis in the parasite. The authors clearly demonstrate that FKBP35 is essential for parasite growth and that ribosome biogenesis is disrupted, but further insights into the pathway itself would be more convincing that this is a direct consequence rather than a secondary feature of parasite death.

The knockdown of PfFKBP35 has no phenotypic consequence, showing that very low amounts of FKBP35 are sufficient for parasite survival and growth. In the absence of quantification of the protein during the course of the experiments, it remains unclear whether the delayed death-like phenotype in the knockout is due to the delayed depletion of the protein or to a delayed consequence of early protein depletion. This limitation also impacts the interpretation of the drug assays.

The authors investigate the activity of FK506 on *P. falciparum*, and conclude that FK506 exerts its antimalarial effects independently of FKBP35, based on the observation that FK506 has the same activity on FKBP35 wild type and knock-out parasites, indicating that FK506 activity is independent of FKBP35 levels. Using cellular thermal shift assays, the authors confirm the interaction between FK506 and FKBP35, and further identify candidate proteins bound by the compound, albeit at lower affinity. Further work is needed to validate whether these putative targets contribute to the FKBP35-independent antimalarial activity of FK506.

---

## [Author Response]

The following is the authors’ response to the previous reviews

We thank the Reviewers and Editors for the evaluation of our revised manuscript.

We especially value the careful assessment of Reviewer 1; at the same time, we clearly disagree with the reviewer’s statement that the revised manuscript “is essentially unchanged”. As appreciated by the other Reviewers, we performed a key experiment (in our opinion the only conclusive experiment) to further solidify that FK506-treatment kills parasites in a FK506-independent manner. Of note, however, Reviewer 1 made us aware of an error in the legend of Figure 4F, which likely contributed to the confusion regarding the antiplasmodial effect of FK506: Unfortunately, we missed updating this legend to appropriately imbed the new experiment. We therefore incorrectly stated that parasites were exposed to FK506 for 48 hours after FK506 treatment at 4-10 hpi and 36-42 hpi in G1. In contrast to the experiments described in the initial submission, parasite survival was not measured 48 h later, but in G2 ring stage parasites, i.e. at a time point during which parasitemia is not affected by the knockout of PfFKBP35. We have now corrected this. As pointed out correctly by Reviewer 1, it would otherwise not be possible to disentangle the effects of the gene knockout and the drug. The setup we now present in Figure 4F, however, is clearly able to do so.

We apologize for the inaccuracy and hope this resolves the ambiguities regarding the FKBP35-independent antimalarial effect of FK506. In line with the comments of Reviewers 2 and 3, we believe that our findings on FK506 activity are of particular importance for the malaria research community. We therefore hope that the final eLife assessment will reflect this.